# Fairness with Wasserstein Adversarial Networks

## Abstract

Quantifying, enforcing and implementing fairness emerged as a major topic in machine learning. We investigate these questions in the context of deep learning. Our main algorithmic and theoretical tool is the computational estimation of similarities between probability, "à la Wasserstein", using adversarial networks. This idea is flexible enough to investigate different fairness constrained learning tasks, which we model by specifying properties of the underlying data generative process. The first setting considers bias in the generative model which should be filtered out. The second model is related to the presence of nuisance variables in the observations producing an unwanted bias for the learning task. For both models, we devise a learning algorithm based on approximation of Wasserstein distances using adversarial networks. We provide formal arguments describing the fairness enforcing properties of these algorithm in relation with the underlying fairness generative processes. Finally we perform experiments, both on synthetic and real world data, to demonstrate empirically the superiority of our approach compared to state of the art fairness algorithms as well as concurrent GAN type adversarial architectures based on Jensen divergence.

**Keywords:** Fair learning, Wasserstein distance, Adversarial Networks.

## 1 Introduction

Along the last few years, much emphasis has been laid on fairness issues in machine learning. Actually, when the learning sample presents biases, these are learnt by algorithms based on loss functions promoting closeness to observed data. Using such models for decision making generalizes biases to the whole population. This drawback of machine learning, also known as unfairness, has become a major challenge in the domain. For a recent survey on this topic we refer to Dwork et al. (2012); Zemel et al. (2013) or Friedler et al. and references therein.

Fairness usually deals with situations where an algorithm exhibits a different behavior for two different subgroups of the population, while these subgroups should not play any role in its outcome. This situation is often modeled as follows : the algorithm should aims at forecasting a variable $Y$ based on observations $X$. Fairness is then defined with respect to a protected variable, called protected attribute, $S$ which represents membership to each population subgroup. The algorithm is called fair if its predictions does not depend too much on $S$.

Defining and quantifying this notion of dependency is a complicated task and has received much attention. One of the main tools is the so-called *disparate impact* which measures if the decision taken by an algorithm differs from one group to another. Absence of disparate impact is called demographic parity. Another measure of fairness is given by the dependency of prediction error with respect to $S$. The independent case is a form of fairness called equality of odds. We refer for instance to Chouldechova (2017), Friedler et al. (2016) or Besse et al. (2018) and references therein. Both situations amounts to considering that either the distribution of the prediction, or its conditional distribution given the target variable $Y$, does not depend on $S$. Hence fairness quantification can be naturally implemented using distance between conditional distributions.

This point of view has been extensively studied when trying to "*repair*" data sets as described for instance in Feldman et al. (2015), Johndrow & Lum (2017), Hacker & Wiedemann (2017) or Friedler et al. (2019). This solution consists in changing the input data so that predictability of the protected

attribute is impossible. The data will be blurred in order to obtain a fair treatment of the protected class. The natural distance to measure the difference between the conditional distributions is the so-called Wasserstein distance, which provides an alternative framework to measure the dependency of the decision rule with respect to the protected attribute as shown in Barrio et al. (2019a) or Barrio et al. (2019b). Yet previous methods face the difficulty of computing the Wasserstein distance which is a challenging task as shown in Peyré & Cuturi (2019).

In this work, we aim at building fair classifiers by considering a Wasserstein type constraint. Adding constraints to the classifiers to get fair behavior has been studied in several papers. We refer to Friedler et al. (2019), Zafar et al. (2017a) and references therein. Our approach, yet sharing some similarities with Edwards & Storkey (2016), based on Ganin et al. (2016), is more flexible and enables to solve wider classes of fairness problems based on different adversarial architecture resulting in more suited loss functions. Wasserstein constraint for fairness has also been considered in Jiang et al. (2019) for binary logistic regression. In the following, we provide algorithms which, for both demographic parity and equality of odds, can incorporate fairness constraints based on the 1-Wasserstein distance. Here we will consider two different mathematical models, describing the relationships between the variables $X$ the target variable $Y$ and the protected variable $S$. Computing Wasserstein type constraints is difficult, we use neural networks as they have been proved useful to estimate Wasserstein type distances as discussed in Arjovsky et al. (2017). The proposed approach can be combined with any kind of neural network predictor. Hence we are able to manage a large variety of input data structure (e.g. images) as well as output labels (multiclass, regression, images . . . ). We demonstrate on fairness benchmark datasets that the proposed Wasserstein approximation framework outperforms both classical fair algorithms (e.g fair SVM) as well as similar adversarial architectures based on Jensen / GAN losses very close to the approaches described in Beutel et al. (2017); Madras et al. (2018).

The paper falls into the following parts. Section 2 is devoted to the presentation of Wasserstein distance, approximation schemes and applications to fair modeling. Section 3 describes a first model of fairness related to demographic parity. Section 4 considers a second option connected to equality of odds. For both models, we propose an adversarial network methodology to obtain a fair classifier for each type of fairness. Section 5 studies these algorithms on real benchmark data sets as well as synthetic simulations.

## 2  FAIRNESS : DEFINITIONS AND METRICS

### 2.1  FRAMEWORK

The statistical model we consider is the following. The problem consists in forecasting a binary variable $Y \in \{0, 1\}$, using observed covariates $X \in \mathbb{R}^d$, $d \geq 1$. We assume moreover that the population can be divided into two categories that represent a bias, modeled by a variable $S \in \{0, 1\}$. This variable is called the protected, or sensitive, attribute which takes the values $S = 0$ for the " minority" class and $S = 1$ for the " majority " class. [1] We observe $n$ joint realizations of these variables $D = \{(X_i, S_i, Y_i), i \in \{1, \ldots, n\}\}$. We use the following notations $D_l = \{(X_i, Y_i), i \in \{1, \ldots, n\}, S_i = l\}$ for $l = 0, 1$, $D_l^k = \{X_i, i \in \{1, \ldots, n\}, S_i = l, Y_i = k\}$ for $l = 0, 1$, $k = 0, 1$.

The *fair* classification problem aims at predicting $Y$ from the variables $X$, using a family of binary classifiers $g \in \mathcal{G} : \mathbb{R}^d \to \{0, 1\}$ without using the information conveyed by $S$. For every $g \in \mathcal{G}$, the outcome of the classification will be the prediction $\hat{Y} = g(X)$.
We consider in the following that the classifier $g$ comes from a score given by a predictor $F : \mathbb{R}^d \to \mathbb{R}$ such that $\hat{Y} = \mathbb{1}_{F(X) > \eta}$ for a chosen threshold $\eta > 0$.

Different criteria have been proposed for measuring the fairness of $\hat{f}$ depending of the context. The **disparate impact (DI)** measures the sensitivity of the predicted values $\hat{Y}$ with respect to $S$.

$$DI(\hat{Y}, S) = |P(\hat{Y} = 1 | S = 0) - P(\hat{Y} = 1 | S = 1)|.$$

---

[1]Note that in the case where $S$ is not a binary variable but multidimensional or multi-class, we can consider one versus one fairness identifying in each case a "minority".

The most favorable situation in terms of fairness with respect to the protected attribute $S$, is achieved when $DI(\hat{Y}, S) = 0$ (i.e. $P(\hat{Y} = 1|S = 0) = P(\hat{Y} = 1|S = 1)$), which corresponds to the situation known as demographic parity. In this case, the predicted class is independent from $S$.

While its interpretation is clear, the mathematical properties of the disparate impact measure are not favorable, in particular it lacks robustness and smoothness features which would be necessary to blend algorithmic practice and mathematical theory. In the following, we propose an alternative measure of equality of opportunity which features smoothness properties and comes with a strong mathematical background. Given a score function $F$, we set $\mathcal{L}_1(F(X)) = \mathcal{L}(F(X)|S = 1)$ and $\mathcal{L}_0(F(X)) = \mathcal{L}(F(X)|S = 0)$ the laws of conditional distribution of the score for each class and denote the corresponding quantile functions by $Q_{0,F}$ and $Q_{1,F}$. Independence of the decision with the variable $S$ would entail that the repartition of the scores is similar for the two subgroups. So the distance between the quantiles of these two distributions acts as a measure of fairness measuring that the repartition of the score is spread in a similar ways whatever the values of the protected attribute, hence acting as a sensitivity index of the predicted values $F(X)$ with respect to $S$. Namely define

$$EMD(F(X), S) := \mathcal{W}(\mathcal{L}_0(F(X)), \mathcal{L}_1(F(X))) = \int_0^1 |Q_{0,f}(t) - Q_{1,f}(t)|dt, \tag{1}$$

which corresponds to the so-called earth-mover or $\mathcal{W}_1$ Wasserstein distance between the conditional distributions. Clearly $\mathcal{W}(\mathcal{L}_0(F(X)), \mathcal{L}_1(F(X))) = 0$ implies that $DI(\hat{Y}, S) = 0$. So Wasserstein distance appears in this framework as a smooth criterion to assess the sensitivity w.r.t to the protected variable. This criterion corresponds to the quantity that is used to measure fairness in Barrio et al. (2019b) and Barrio et al. (2019a). Note that Wasserstein distance for fairness has been also considered in the seminal paper by Feldman et al. (2015).

Another important criterion is the *equality of odds*, which measures the influence of the $S$ on the accuracy of the algorithm. For this the prediction errors across the different class groups are compared and this notion of fairness is achieved when $P(\hat{Y} = 1|Y = 1, S = 0) = P(\hat{Y} = 1|Y = 1, S = 1)$ and $P(\hat{Y} = 0|Y = 0, S = 0) = P(\hat{Y} = 0|Y = 0, S = 1)$. Here again, this condition can be interpreted as a notion of independence of the conditional distributions defined for $(i, s) \in \{0, 1\}^2$ as $\mathcal{L}_s^i(f(X))$ the distribution of the random variable $(f(X)|Y = i, S = s)$. Hence as we exposed for the notion of *equality of opportunity*, fairness will be assessed through the computations of the Wasserstein distances $\mathcal{W}(\mathcal{L}_0^0(F(X)), \mathcal{L}_1^0(F(X)))$ and $\mathcal{W}(\mathcal{L}_0^1(F(X)), \mathcal{L}_1^1(F(X)))$.

Note that in some cases, we are only interested in equality of opportunities. This corresponds to the case where we only require that $P(\hat{Y} = 1|Y = 1, S = 0) = P(\hat{Y} = 1|Y = 1, S = 1)$ as pointed out in Hardt et al. (2016). Hence in this case it amounts to control only $\mathcal{W}(\mathcal{L}_0^1(F(X)), \mathcal{L}_1^1(F(X)))$.

## 2.2 WASSERSTEIN DIVERGENCES USING NEURAL NETWORKS AND PROPERTIES

The earth-mover, or Wasserstein-1 distances between probability distribution is defined as follows :

$$\mathcal{W}(\mathcal{L}_1, \mathcal{L}_2) = \inf_{\gamma \in \Pi(\mathcal{L}_1, \mathcal{L}_2)} \mathop{\mathbb{E}}_{X, Y \sim \gamma} \parallel X - Y \parallel \tag{2}$$

where $\Pi(\mathcal{L}_1, \mathcal{L}_2)$ is the set of all probability measures on $X, Y$ with marginals $\mathcal{L}_1$ and $\mathcal{L}_2$. Disparate Impact is closely related to the notion of unpredictability of the variable $S$. Hence the aim in this case is to The distance associated to these notions is the total variation distance $d_{\text{TV}}(\mathcal{L}_1, \mathcal{L}_2$ but due to intractability of this distance, it has been replaced in the machine learning literature by $\mathcal{W}$. Clearly the independent case is obtained when the distance is null and the decrease of $\mathcal{W}$ leads to smaller DI as shown in the experiments in Barrio et al. (2019a). Hence a constraint on the Wasserstein distance promotes fairness, also in terms of Disparate Impact.

Although the infimum in Equation (2) is not tractable in general, it can be approximated by a neural network. The first step is to reformulate (2) using the Kantorovich-Rubinstein duality Villani (2008):

$$\mathcal{W}(\mathcal{L}_1, \mathcal{L}_2) = \sup_{f \in \mathcal{F}_1} \mathbb{E}_{X \sim \mathcal{L}_1} f(X) - \mathbb{E}_{X \sim \mathcal{L}_2} f(X) \tag{3}$$

where $\mathcal{F}_1$ denotes the space of 1 Lipschitz function. As a second step, the approach proposed in Arjovsky et al. (2017) is based on estimation of the supremum in (3) by replacing $\mathcal{F}_1$ by the set of functions described by a fixed neural network architecture with spectral normalization Miyato et al.

(2018). This provides a general methodology to estimate and optimize divergences *à la* Wasserstein and leads to interesting empirical results Arjovsky et al. (2017). Furthermore, we demonstrate empirically that this approach allows to control to some extent the empirical EMD divergence introduced above. The two examples of fairness measures which we have introduced are based on distributional divergence measured using Wasserstein distance and we propose to handle these divergence terms computationally using the dual formulation presented in this paragraph.

One specificity of the fairness problem which we consider is that, empirically, we only have access to a finite number of samples for each values of the protected attribute ($S \in \{0, 1\}$). For example, we only have access to $\mathcal{L}_0$ and $\mathcal{L}_1$ through a fixed finite sample and the Wasserstein terms which we manipulate are only computed on finitely many samples. This raises the following comments.

Other divergences, such as Jensen-Shannon Goodfellow et al. (2014), KL divergence or total variation Arjovsky et al. (2017) can be approximated using neural networks. These divergences reflect similarities between mutually absolutely continuous probability measures. However they degenerate when considering singular measures. For the problems which we intend to attack in this work, we aim at enforcing equality of distributions using only a fixed number of samples. Entropy or total variation based divergences fail to capture dissimilarity between singular measures, and in particular they degenerate when considering disjoint finite sample sets. On the other hand, Wasserstein metric is well defined and does not degenerate on empirical distributions given by finite samples.

A second favorable property of this metric is its continuity features. When considering parametric distributions, this translate into continuity of the metric in the parameter space as remarked by Arjovsky et al. (2017), resulting in numerically more favorable situations compared to discontinuous problems. Another important consequence of the continuity properties of Wasserstein distance is that it translates into stable approximation of distribution divergence in the limit of large *i.i.d* samples (see Appendix). This last property is very desirable since all we can do from an empirical perspective is limited to finite samples.

## 3 TYPE 1 FAIR LEARNING : DEMOGRAPHIC PARITY

The first case where fairness is desirable corresponds to the situation where the target variable is biased (....). For instance, it is well-known that the income of a people is biased by the gender. The situation doesn't arise from a biased gathering of data but from bias that exist in the real data and that we don't want to reproduce in our model. Thus, a fair model in this case will change the prediction in order to make them independent from the protected variable. A suitable objective for this problem is to obtain of disparate impact (or the SDI) as close as possible to 1.

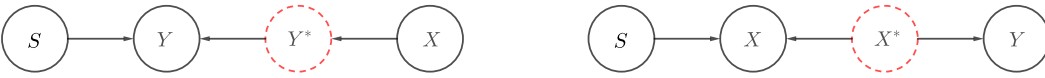

Figure 1: Fairness type 1.         Figure 2: Fairness type 2.

This situation is formally represented in Figure 1. In this situation we require that :

$$X \perp\!\!\!\perp S | Y \qquad \text{and} \qquad Y^* \perp\!\!\!\perp S | Y$$

where $Y^*$ is not observed. Note that, as it intuitively expected, $Y$ is not independent from $S$ (even conditionally to $X$). In this example, $Y^*$ could represent an ideal case where the income level reflect the proficiency and not the gender.

### 3.1 TYPE 1 FAIR NEURAL NETWORKS

The following can be applied either in multivariate regression, $Y \in \mathbb{R}^d$, or classification $Y = \{0, 1\}$, we consider the type 1 configuration described in Figure 1. We propose a neural network model with adversarial Wasserstein constraints on the output as described in Figure 3 In this networks, the function $F$ is a classifier or regressor. In order to have a prediction independent from $S$, we add

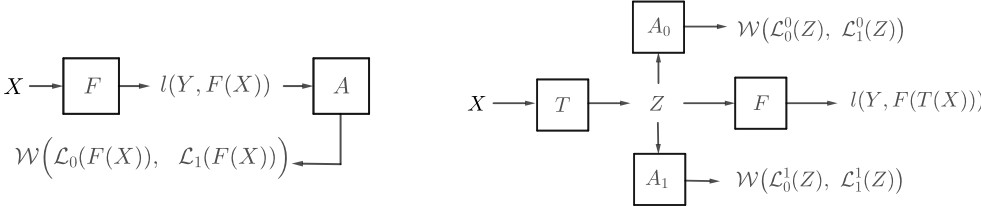

Figure 3: Networks for fairness type 1

Figure 4: Fair networks for fairness type 2

---

**Algorithm 1** Type 1 Fair learning algorithm

---

**Require:** $\alpha$, the learning rate, $\lambda$ the fairness constraint, $m$ the batch size, $nb\_iter$, the number of iterations, $n_w$ the number of iterations for the Wasserstein estimators.

1: **for** $k$ in $nb\_iter$ **do**
2:     **for** $j$ in $n_w$ **do**
3:         sample *iid* $\{x_i, y_i\}_{i=1}^m \sim \mathcal{L}(D_0)$, and $\{x_i', y_i'\}_{i=1}^m \sim \mathcal{L}(D_1)$
4:         update $A$ with an by gradient ascent : $\nabla_{A_0} \frac{1}{m} \sum_{i=1}^m \big( A(F(x_i)) - A(F(x_i')) \big)$
5:         Normalize $A$ as in Miyato et al. (2018).
6:     **end for**
7:     sample *iid*, $\hat{D}_k = \{d_i = (x_i, s_i, y_i)\}_{i=1}^m \sim \mathcal{L}(D)$
8:     update $F$ by gradient descent :

$$\nabla_F \frac{1}{m} \sum_{i=1}^m l(F(x_i), y_i) + \frac{1}{|\hat{D}_k \cap D_0|} \sum_{d_i \in D_0} A(F(x_i)) - \frac{1}{|\hat{D}_k \cap D_1|} \sum_{d_i \in D_1} A(F(x_i))$$

9: **end for**

---

Wasserstein penalization reflecting the dependency between $F(X)$ and $S$, we obtain the following optimization:

$$\inf_F \mathbb{E}_X[l(F(X), Y)] + \lambda \mathcal{W}(\mathcal{L}_0(F(X)), \mathcal{L}_1(F(X))) \tag{4}$$

where $l$ is the loss function for the problem. Note that the Wasserstein penalty term in (4) is exactly the $EMD$ fairness measure which we introduced in (1). Applying the approximation scheme described in Section 2.2, we obtain the following saddle point problem :

$$\inf_F \sup_{A \in \mathcal{F}_s} \mathbb{E}_X[l(F(X), Y)] + \lambda \big( \mathbb{E}_{X \sim \mathcal{L}_0}[A(F(X))] - \mathbb{E}_{X \sim \mathcal{L}_1}[A(F(X))] \big)$$

$\lambda > 0$ are hyper-parameters and $\mathcal{F}_s$ represents the set of functions encoded by a fixed architecture neural network with spectral normalization. Approximating expectations using empirical sample, the learning process for this model is described in Alg. 1. As explained in Section 2.2, in the limit of large samples, the maximization in $A$ provide a proxy for the Wasserstein distance between the two conditional distributions. Note that we can use a similar architecture for equality of opportunities.

## 4 TYPE 2 FAIR LEARNING : EQUALITY OF ODDS

The second case where fairness is desirable corresponds to the situation where the data are subject to a bias nuisance variable which is in principle of no help for the learning task at hand and which influence should be removed. On famous example is the dog vs wolf problem exposed in Ribeiro et al. (2016). In this example, that data was heavily biased by the presence, for the wolfs, and the absence, for the dogs, of snow in the picture. Although the presence of snow is not independent from the presence of wolfs, we prefer a model that focuses on animal features rather than background. More generally, this kind of situation appears when the descriptors or target variables show dependency with the protected variable due to a biased data collection process or when we plan to use the model on data that have a different distribution with respect to the protected variable (this could be the case

if we want to detect wolfs and dogs in a snow free area). The equality of odds is a suitable objective for this type of fairness.

We represent the underlying data generation process formally in Figure 3, we require the following conditional independence:

$$X^* \perp\!\!\!\perp S|Y, \qquad Y \perp\!\!\!\perp S|X^*$$

where $X^*$ is not observed. Note that, as it is intuitively expected, $Y$ and $X$ are not independent from $S$ (even conditionally on any other variable in the model). In this context, we suppose that there is a representation of the data $X^*$ from which we can build a model to predict $Y$ which will be independent of $S$ given $Y$. Back to our example for wolf and dog, $X$ could be the pictures, $X^*$ could be physical features of the animal. A model learnt from this $X^*$ could predict $Y = \{dog, wolf\}$ from a picture independently of the presence of snow (even if the probability of observing snow is greater when the animal on a picture is a wolf).

### 4.1 Type 2 fair neural networks for binary classification

In the following, we consider the case where $Y \in \{0,1\}$ and type 2 configuration. We propose a neural network model with adversarial Wasserstein constraints as described in Figure 3.1

In this networks, the function $F \circ T(X)$ (or $F(Z)$ with $Z = T(X)$) is a classifier constructed in two steps : a transformation $T : X \to Z$ and a classifier $F : Z \mapsto F(Z) \in 0, 1$. We expect to build $T$ such that $Z$ has the same properties as $X^*$ (i.e. $X^* \perp\!\!\!\perp S|Y$). In order to achieve this goal, we constraint the distribution $Z$ conditionally to $Y$ to be independent of $S$. Based on this idea, we obtain the following optimization problem :

$$\inf_{F,T} \quad \mathbb{E}_X \left[ l(F(T(X)), Y) \right] + \lambda \left[ \mathcal{W}(\mathcal{L}_0^0(T(X)), \mathcal{L}_1^0(T(X))) + \mathcal{W}(\mathcal{L}_0^1(T(X)), \mathcal{L}_1^1(T(X))) \right] \quad (5)$$

where $l$ is a given loss function (binary cross entropy in our setting). We then apply the approximation procedure of Wasserstein distance described in Section 2.2 and obtain the following saddle point problem :

$$\inf_{F,T} \quad \mathbb{E}_X \left[ l(F(T(X)), Y) \right]$$
$$+ \lambda \sup_{A_0, A_1 \in \mathcal{F}_s} \underset{\mathcal{L}_0^0(X)}{\mathbb{E}} A_0(T(X)) - \underset{\mathcal{L}_1^0(X)}{\mathbb{E}} A_0(T(X)) + \underset{\mathcal{L}_0^1(X)}{\mathbb{E}} A_1(T(X)) - \underset{\mathcal{L}_1^1(X)}{\mathbb{E}} A_1(T(X))$$

where $\lambda > 0$ are hyper-parameters of the method and $\mathcal{F}_s$ describes all functions generated by a given fixed architecture neural network with spectral normalization Miyato et al. (2018). Based on finite sample approximation of the various expectations in this formulation, the learning process is similar to Alg. 1 and is fully described in the appendix. The supremum over $A_0$ and $A_1$ for finite sample approximation of the expectations is a proxy to the Wasserstein distance between the two conditional distributions of interest in 2.1. Moreover, Property 1 states that in the limit of Wasserstein distance between conditional distribution set to 0 the latent space $Z = T(X)$ satisfies the same properties of conditional Independence as $X^*$. Furthermore any classifier build *a posteriori* on $Z$ will satisfy equal opportunities with respect to $Y$.

**Proposition 1** *Assume that the deterministic map $T$ satisfies*

$$\mathcal{W}\left( \mathcal{L}_0^0(T(X)), \mathcal{L}_0^1(T(X)) \right) = 0 \qquad and \qquad \mathcal{W}\left( \mathcal{L}_0^1(T(X)), \mathcal{L}_1^1(T(X)) \right) = 0$$

*then, we have $T(X) \perp\!\!\!\perp S|Y$, for any measurable map $G$, $G(T(X)) \perp\!\!\!\perp S|Y$.*

**Proof sketch:** Both are expressions of equality in distribution. Nullity of Wasserstein distance entails $T(X) \perp\!\!\!\perp S|Y$. This implies that for any deterministic map $G$, $G(T(X)) \perp\!\!\!\perp S|Y$. $\qquad \square$
Note that, contrary to other repair procedures for which the transformation must be recomputed for any new observations (in Barrio et al. (2019a) the transformation relies on the optimal transport map which depends on the observations), here the optimal transformation $T$ can be used directly for all new observations.

Table 1: Accuracy, Disparate impact and EMD for adult, bank and CelebA under demographic parity and equality of odds constraint

| Type 1 : demographic parity | | | | | | | | | |
|---|---|---|---|---|---|---|---|---|---|
| data | Adult gender | | | Bank | | | CelebA | | |
| score  Alg | ACC | DI | EMD | ACC | DI | EMD | ACC | DI | EMD |
| UnfairClf | 84.5[0.5] | 0.19 | 0.18 | 90.3[0.2] | 0.14 | 0.16 [0.7] | 77.9 [0.7] | 0.46 | 0.38 |
| C-SVM | 79.4 [0.6] | 0.02 | - | 90.0 [0.2] | 0.02 | - | - | - | - |
| Wass. Alg1 $\lambda = 0.9$ | 80.7 [0.7] | 0.01 | 0.02 | 90.0 [0.2] | 0.01 | 0.01 | 71.2 [0.1] | 0.06 | 0.05 |
| Gan. Alg1 $\lambda = 3$ | 80.3 [0.7] | 0.03 | 0.04 | 87.2 [0.9] | 0.07 | 0.05 | 74.1 [0.3] | 0.18 | 0.17 |
| Wass. Alg1 $\lambda = 0.65$ | 83.2 [0.5] | 0.07 | 0.03 | 90.2 [0.2] | 0.01 | 0.02 | 74.6 [0.2] | 0.18 | 0.17 |
| Gan. Alg1 $\lambda = 1$ | 83.9 [0.4] | 0.09 | 0.09 | 90.4 [0.2] | 0.09 | 0.08 | 75.4 [0.3] | 0.23 | 0.22 |
| Type 2 : equality of odds | | | | | | | | | |
| | ACC | $DI_Y$ | $EMD_Y$ | ACC | $DI_Y$ | $EMD_Y$ | ACC | $DI_Y$ | $EMD_Y$ |
| UnfairClf | 84.5[0.5] | 0.19 | 0.18 | 89.9[0.5] | 0.07 | 0.08 | 78.1 [0.7] | 0.48 | 0.38 |
| Wass. Alg1 $\lambda = 1$ | 81.0 [0.6] | 0.03 | 0.05 | 89.2 [0.2] | 0.04 | 0.05 | 77.2 [0.7] | 0.27 | 0.23 |
| Gan. Alg1 $\lambda = 2$ | 81.9 [0.7] | 0.09 | 0.08 | 89.1 [0.2] | 0.04 | 0.05 | 77.1 [0.6] | 0.27 | 0.23 |

## 5 EXPERIMENTATION

We show in this section, empirical results supporting our theoretical expectations. For simplicity and reproducibility purposes, we keep neural networks as simple as possible and try to use similar architectures as much as we can. For all the experiments, we set the learning rate of Adam to $1e^{-4}$, and $n_w$ to 10 (see algorithm description in Sections 3 and 4). All experiments have been implemented with keras/tensorflow.

**Fairness benchmarks:** We consider three state of the art fairness benchmarks : (i) impact of gender in adult database (predict income>50K, 48842 examples, 16 attributes) (ii) impact of age (boolean $25 < age > 60$) in the bank database (predict credit acceptance, 45211 examples, 17 attributes) (iii) impact of the gender on the attractivity in a subset of the celebA dataset (64x64 rgb images, 19670 examples).

**Concurrent methods:** We compare our results with the C-SVM implementation proposed by Zafar et al. (2017a;b) and a classifier based on our neural network architecture without fairness constraint (UnfairClf). We also compare our approach with GAN type Jensen adversarial as in Beutel et al. (2017); Madras et al. (2018). Note that the architectures that we use are slightly different from the original papers, this was on purpose to ensure an objective comparison with our approach. Indeed we keep our adversarial architectures and only replace Wasserstein loss and network by a classifier with binary cross entropy and GAN trick for training.

**Results:** Table 1 reports accuracy (ACC), DI and EMD in a 70%train-30%test scheme with 10 repetitions to assess variability[2] in both demographic parity and equality of odds scenarios. For equality of odds, we aggregate fairness measures $DI$ and $EMD$, conditioning on $Y$ and summing over $Y = 0, 1$, these aggregated measures are denoted by $DI_Y$ and $EMD_Y$. For the adversarial approaches we report one result with high fairness constraint and one result with a lesser constraint (obtained by considering different values of $\lambda$). For the demographic parity constraint, in all examples, our algorithm reduces the DI close to 0 with an acceptable accuracy decrease. Figure 5 illustrates that our last hyper-parameter $\lambda$ (see Equation (4)) controls the trade-off between accuracy and fairness both in wasserstein and GAN configurations. Our approach clearly dominates concurrent methods in all situations in terms of both accuracy and fairness level. This is further illustrated in the third part of Figure 5 where the accuracy / fairness tradeoff is very favorable to our wasserstein approach compared to more traditional GAN methods. Finally Table 1 illustrates difficulties for GAN models to enforce hard fairness constraints (DI close to 0). For equality of odds, our method and GAN approach perform similarly.

---

[2]Standard deviation is only reported for accuracy because it was negligible for other quantities

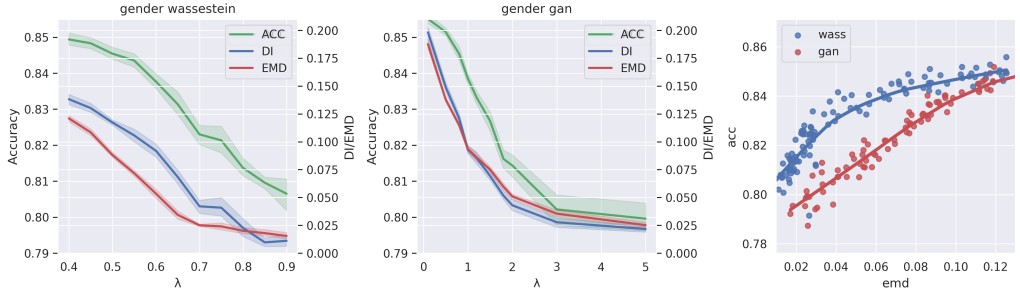

Figure 5: Accuracy / fairness tradeoffs between our Wasserstein approach and more traditional GAN approaches similar to Beutel et al. (2017); Madras et al. (2018) for demographic parity.

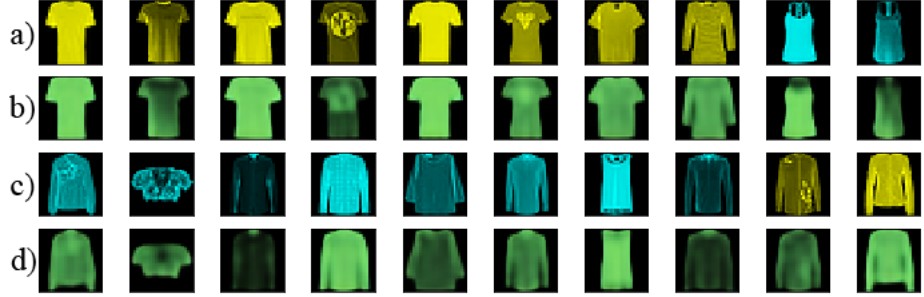

Figure 6: Fair auto encoder.a) T-shirt, b) fair representation of T-shirt, c) shirt, d)fair representation of shirt

**Learning based on fair representations:** To illustrate Proposition 1 and the fact that our fairness constrinat can be applied to other type of problem than classification, we consider classification of T-shirts versus shirts ($Y = \{Tshirt, shirt\}$) in the fashion-MNIST dataset Xiao et al. (2017) (12000 training examples, 2000 test examples). These two classes are known to be the most challenging to distinguish in this dataset (accuracy around 0.9). We bias the dataset by adding a color ($S = \{turquoise, yellow\}$) correlated to the class variable $Y$: $P(S = yellow|Y = T - shirt) = 0.9$, $P(S = Turquoise|Y = shirt) = 0.9$. We apply the following experimental process: train on the biased dataset, and compare validation performances both on the biased test set and the same biased test set, with switched colors: $P(S = yellow|Y = T - shirt) = 0.1$ and $P(S = Turquoise|Y = shirt) = 0.1$.

We train a fair auto-encoder with two Wasserstein adversarial networks constraining equality of odds for the decoded images. We observe in Figure 5 that equality of odds is achieved by assigning the same color to all transformed images. We train a first network (unfairClf) on the biased database. We train a similar network on the fair database, and construct a fair classifier (fairClf) by composition of the second trained classifier and the auto-encoder. As expected unfairClf generalizes better on the biased test set (accuracy 0.94 versus 0.87). However when switching the test color distribution fairClf is far more robust (accuracy 0.82 versus 0.60). This demonstrates that in addition to building a representation which looks fair, our auto-encoder approach is robust to fluctuations of the bias variable distribution.

## 6 CONCLUSIONS

This work tackles the challenge of incorporating constraints to deal with bias issues in machine learning. We show that Wasserstein is an appropriate choice of distance between conditional distributions to control fairness using adversarial neural networks. We also explicit mathematical models providing abstract frameworks to understand and apply two types of fair constraints (demographic parity and equality of odds). The predictor we obtain prove efficient on well known fairness benchmarks

as well as synthetic problems. Our experiments designed with minimal hand tuning to overcome reproducibility issues. As expected adversarial wasserstein constraints are more efficient to enforce fairness than their traditional GAN counterparts.

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

This is supplementary file for the paper *Fairness with Wasserstein Adversarial Networks*.

## A    SUPPLEMENTARY

### A.1    CENTRAL LIMIT THEOREM FOR EMPIRICAL WASSERSTEIN IN DIMENSION 1

The behavior of the empirical transportation cost is a key issue and heavily relies on the dimension of the problem. For general dimension we refer to Dobrić & Yukich (1995) for the asymptotic limit of the transportation costs. The asymptotic behavior is provided for quadratic cost in general dimension in del Barrio & Loubes (2019). For the one dimensional case, which corresponds to the situation encountered here since the constraint is imposed on a score $F(X) \in \mathbb{R}$, it is possible to control more precisely for general costs the asymptotic behavior of the empirical Wasserstein's distance. The following theorem is an extension for the two sample case and $\mathcal{W}$ of the theorem found in Barrio et al. (2019b). We provide it for sake of completeness.

**Theorem 1** *Assume that $F$ and $G$ satisfies the following assumption*

$$\int_{-\infty}^{\infty} \sqrt{F(t)(1 - F(t))}dt < \infty \tag{6}$$

*If $\lambda(F = G) = 0$ where $\lambda$ is Lebesgue measure on $\mathbb{R}$. Assume that $\frac{n}{n+m} \to \pi \in (0, 1)$. Then*

$$\sqrt{\frac{nm}{n+m}}\big((\mathcal{W}_1(F_n, G_m) - \mathcal{W}_1(F, G)\big) \to_w (1 - \pi)N(0, \sigma_1^2(F, G)) + \pi N(0, \sigma_1^2(G, F)),$$

*with $\sigma_1^2(F, G) = \int_0^1 c_1^2(t; F, G)dt - \left( \int_0^1 c_1(t; F, G)dt \right)^2$ and*

$$c_1(t; F, G) := \int_{F^{-1}(\frac{1}{2})}^{F^{-1}(t)} \operatorname{sgn}\big(s - G^{-1}(F(s))\big)ds, \quad 0 < t < 1.$$

**Proof sketch:**    This results follows the proofs in Barrio et al. (2019b). It can be deduced from the one sample case which states that

$$\sqrt{n}\big(\mathcal{W}_1(F_n, G) - \mathcal{W}_1(F, G)\big) \to_w \int_{\mathbb{R}} v_F(x)dx,$$

where $v_F(x) = B(F(x))$ if $F(x) > G(x)$, $v_F(x) = -B(F(x))$ if $F(x) < G(x)$, $v_F(x) = |B(F(x))|$ if $F(x) = G(x)$ and $B$ is a Brownian bridge on $[0, 1]$. In particular, then

$$\sqrt{n}\big(\mathcal{W}_1(F_n, G) - \mathcal{W}_1(F, G)\big) \to_w N(0, \sigma_1^2(F, G)),$$

Note that Wasserstein distance is expressed through the quantile functions but when $p = 1$, we have by simple arguments that

$$\mathcal{W}(P_n, Q) = \int_{\mathbb{R}} |F_n(x) - G(x)|dx,$$

which allows to deal with the empirical transportation cost through the consideration of the process

$$\alpha_n^F(x) := \sqrt{n}(F_n(x) - F(x)), \quad x \in \mathbb{R}.$$

Under the assumption

$$\int_{-\infty}^{\infty} \sqrt{F(t)(1 - F(t))}dt < \infty$$

we have that $\alpha_n^F$ converges weakly in $L_1(\mathbb{R})$ to $B^F$, a centered Gaussian process on $\mathbb{R}$ with covariance function

$$\operatorname{Cov}\left(B^F(x), B^F(y)\right) = F(x \wedge y) - F(x)F(y),$$

see Theorem 2.1 in del Barrio et al. (1999). By the Skorohod-Dudley-Wichura Theorem (see, e.g., Theorem 11.7.2 in Dudley (2002)), we can, therefore, consider versions of $\alpha_n^F$ and $B^F$ such that $\|\alpha_n^F - B^F\|_{L_1} \to 0$ a.s.. Now,

$$\sqrt{n}\big(\mathcal{W}_1(F_n, G) - \mathcal{W}_1(F, G)\big) = \int_{\mathbb{R}} u_n(x)dx,$$

where $u_n(x) = \sqrt{n}\big(|F(x) - G(x) + \alpha_n^F(x)/\sqrt{n}| - |F(x) - G(x)|\big)$. We introduce $v_n(x) = \sqrt{n}\big(|F(x) - G(x) + B^F(x)/\sqrt{n}| - |F(x) - G(x)|\big)$ and $v(x) = B^F(x)$ is $F(x) > G(x)$, $v(x) = -B^F(x)$ if $F(x) < G(x)$ and $v(x) = |B^F(x)|$ if $F(x) = G(x)$. We note that $|u_n(x) - v_n(x)| \leq |\alpha_n^F(x) - B^F(x)|$, which implies that

$$\left| \int_{\mathbb{R}} u_n(x)dx - \int_{\mathbb{R}} v_n(x)dx \right| \leq \|\alpha_n^F - B^F\|_{L_1} \to 0 \tag{7}$$

with probability one.

Now, if $F(x) > G(x)$ then $v_n(x)$ will eventually equal $B^F(x)$, while if $F(x) < G(x)$ then $v_n(x) = -B^F(x)$ for large enough $n$. Hence, $v_n(x) \to v(x)$ pointwise. On the other hand,

$$|v_n(x)| \leq |B^F(x)|.$$

This shows that we can apply dominated convergence to conclude that

$$\int_{\mathbb{R}} v_n(x)dx \to \int_{\mathbb{R}} v(x)dx. \tag{8}$$

Combining (7) and (8) we see that $\sqrt{n}(\mathcal{W}_1(F_n, G) - \mathcal{W}_1(F, G)) \to \int_{\mathbb{R}} v(x)dx$. To conclude we note that $B^F$ has the same distribution as $B(F(\cdot))$ with $B$ a standard Brownian bridge on $[0, 1]$. Normality and the expression for the variance when $\lambda(F = G) = 0$ follow from the fact that, in that case,

$$\int_{\mathbb{R}} v(x)dx = \int_{\mathbb{R}} B(F(x))h(x)dx$$

with $h(x) = I(F(x) > G(x)) - I(F(x) < G(x))$. This last integral is a centered Gaussian r.v. with variance

$$\int_{\mathbb{R}^2} (F(x \wedge y) - F(x)F(y))h(x)h(y)dxdy = \int_0^1 H^2(t)dt - \left( \int_0^1 H(t)dt \right)^2,$$

where $H(t) = \int_{F^{-1}(\frac{1}{2})}^{F^{-1}(t)} h(s)ds$ (the last equality follows, from instance, from Proposition 7.4.2, p. 117 in Shorack (2000)). Finally, we note that $F(x) > G(x)$ if and only if $G^{-1}(F(x)) > x$ and also that $x = G^{-1}(F(x))$ if and only if $G(x) \geq F(x)$ and $G(y) < F(x)$ for every $y < x$. But then $G(x) = F(x)$ unless $G$ is not continuous at $x$. But this can happen at most for a countable collection of $x$. This means that $I(F(x) > G(x)) = I(G^{-1}(F(x)) > x)$ and, under the assumption $\lambda(F = G) = 0$, that $I(F(x) < G(x)) = I(G^{-1}(F(x)) < x)$ for a.e. $x$. This completes the proof.

## A.2 Asymptotics in the multivariate setting

**Proposition 2** *Given $X_1, \ldots, X_n \sim \mathcal{L}_1$ and $X'_1, \ldots, X'_{n'} \sim \mathcal{L}_2$ be two iid sample on $\mathbb{R}^d$ of compactly supported distributions $\mathcal{L}_1$ and $\mathcal{L}_2$. Then we have*

$$\sup_{f \in \mathcal{F}_1} \frac{1}{n} \sum_{i=1}^n f(X_i) - \frac{1}{n'} \sum_{j=1}^{n'} f(X'_j) \quad \underset{n \to \infty, n' \to \infty}{\longrightarrow} \quad \mathcal{W}(\mathcal{L}_1, \mathcal{L}_2).$$

**Proof sketch:** Let $\mathcal{L}_{1,n}$ and $\mathcal{L}_{2,n'}$ be the two empirical measures associated to our two samples. From the law of large numbers, $\mathcal{L}_{1,n} \underset{n \to \infty}{\to} \mathcal{L}_1$ and $\mathcal{L}_{2,n'} \underset{n' \to \infty}{\to} \mathcal{L}_2$ where the limit is understood as convergence in distribution of probability measures (or weak convergence of probability distributions). The result follows combining the equivalence between (2) and (3) with the fact that $\mathcal{W}$ metrizes this notion of convergence (Villani, 2008, Corollary 6.9).

## B Algorithm

## C Architectures

### C.1 General process

We use RELU for all the hidden layers, and implement neither dropout nor batch normalization (except for the celebA classifier which use batch normalization).

---

**Algorithm 2** Type 2 Fair learning algorithm

---

**Require:** $\alpha$, the learning rate, $\lambda$ the fairness constraint, $m$ the batch size, $nb\_iter$, the number of iteration, $n_w$ the number of iterations for the Wasserstein estimators.

1: **for** $k$ in $nb\_epoch$ **do**
2:     **for** $j$ in $n_w$ **do**
3:         sample *iid* $\{x_i\}_{i=1}^m \sim \mathcal{L}_0^0(X)$, and $\{x_i'\}_{i=1}^m \sim \mathcal{L}_1^0(X)$
4:         update $A_0$ by gradient ascent : $\nabla_{A_0} \frac{1}{m} \sum_{i=1}^m \left( A_0(T(x_i)) - A_0(T(x_i')) \right)$
5:         Scale $A$ as in Miyato et al. (2018)
6:         repeat for $A_1$ and the corresponding sampling
7:     **end for**
8:     sample *iid* $\hat{D}_k = \{d_i = (x_i, s_i, y_i)\}_{i=1}^m \sim \mathcal{L}(D)$
9:     update $T$ and $F$ an by gradient descent :

$$\nabla_{F,T} \frac{1}{m} \sum_{i=1}^m l(F(T(x_i), y_i) + \frac{1}{|\hat{D}_k \cap D_0^0|} \sum_{d_i \in D_0^0} A_0(T(x_i)) - \frac{1}{|\hat{D}_k \cap D_1^0|} \sum_{d_i \in D_1^0} A_0(T(x_i))$$

$$+ \frac{1}{|\hat{D}_k \cap D_0^1|} \sum_{d_i \in D_0^1} A_1(T(x_i)) - \frac{1}{|\hat{D}_k \cap D_1^1|} \sum_{d_i \in D_1^1} A_1(T(x_i)) \quad (9)$$

10: **end for**

---

### C.2 TYPE 1 ARCHITECTURES

We present the architecture of the network $F$ for the unfairclf, Type 1 and type 2 architectures used for the Adult and bank dataset :

| ID | Layer | Kernel | Stride | Activation | Normalization | Output |
|----|-------|--------|--------|------------|---------------|--------|
| Input0 | Input | - | - | None | None | nb variables |
| dense1 | dense | - | - | RELU | None | $128 \times 1$ |
| dense2 | dense | - | - | RELU | None | $64 \times 1$ |
| dense3 | dense | - | - | RELU | None | $32 \times 1$ |
| OutputF | Dense | - | - | sigmoid | None | 1 |

We present the architecture of the network $F$ for the unfairclf, Type 1 and type 2 architectures used for the CelebA dataset

| ID | Layer | Kernel | Stride | Activation | Normalization | Output |
|----|-------|--------|--------|------------|---------------|--------|
| Input0 | Input | - | - | None | None | $64 \times 64 \times 3$ |
| Conv1 | Convolutional | $3 \times 3$ | - | RELU | True | $64 \times 64 \times 32$ |
| Conv2 | Convolutional | $3 \times 3$ | - | RELU | True | $64 \times 64 \times 32$ |
| pool0 | max pooling | - | - | - | None | $32 \times 32 \times 32$ |
| Conv3 | Convolutional | $3 \times 3$ | - | RELU | True | $32 \times 32 \times 64$ |
| Conv4 | Convolutional | $3 \times 3$ | - | RELU | True | $32 \times 32 \times 64$ |
| Conv5 | Convolutional | $3 \times 3$ | - | RELU | True | $32 \times 32 \times 64$ |
| pool1 | max pooling | - | - | - | None | $16 \times 16 \times 64$ |
| Conv6 | Convolutional | $3 \times 3$ | - | RELU | True | $16 \times 16 \times 128$ |
| Conv7 | Convolutional | $3 \times 3$ | - | RELU | True | $16 \times 16 \times 128$ |
| Conv8 | Convolutional | $3 \times 3$ | - | RELU | True | $316 \times 16 \times 128$ |
| pool2 | max pooling | - | - | - | None | $8 \times 8 \times 128$ |
| Conv9 | Convolutional | $3 \times 3$ | - | RELU | True | $8 \times 8 \times 256$ |
| Conv10 | Convolutional | $3 \times 3$ | - | RELU | True | $8 \times 8 \times 256$ |
| Conv11 | Convolutional | $3 \times 3$ | - | RELU | True | $8 \times 8 \times 256$ |
| pool3 | max pooling | - | - | - | None | $4 \times 4 \times 256$ |
| Flat 0 | flatten | - | - | - | None | $4096 \times 1$ |
| dense1 | dense | - | - | RELU | None | $32 \times 1$ |
| OutputF | Dense | - | - | sigmoid | None | 1 |

All the wasserstein and GAN constraint networks use the following for the Type 1 architectures :

| ID | Layer | Kernel | Stride | Activation | Normalization | Output |
|---|---|---|---|---|---|---|
| Input0 | Input | - | - | None | None | 1 |
| dense1 | dense | - | - | RELU | spectral for wass | $64 \times 1$ |
| dense2 | dense | - | - | RELU | spectral for wass | $32 \times 1$ |
| OutputA | Dense | - | - | linear (signoid for GAN) | spectral for wass | 1 |

The constraint are always added on the output layer of the $F$ network.

## C.3 FAIR AUTO ENCODER

We present the architecture of the network $T$ for the fair auto-encoder

| ID | Layer | Kernel | Stride | Activation | Normalization | Output |
|---|---|---|---|---|---|---|
| Input0 | Input | - | - | None | None | $28 \times 28 \times 3$ |
| Conv0 | Convolutional | $3 \times 3$ | - | RELU | None | $28 \times 28 \times 64$ |
| pool0 | max pooling | - | - | - | None | $14 \times 14 \times 32$ |
| Conv1 | Convolutional | $3 \times 3$ | - | RELU | None | $14 \times 14 \times 128$ |
| Conv2 | Convolutional | $3 \times 3$ | - | RELU | None | $14 \times 14 \times 256$ |
| pool2 | max pooling | - | - | - | None | $7 \times 7 \times 256$ |
| Conv3 | Convolutional | $3 \times 3$ | - | RELU | None | $7 \times 7 \times 256$ |
| up0 | upsampling | - | - | - | None | $14 \times 14 \times 256$ |
| Conv4 | Convolutional | $3 \times 3$ | - | RELU | None | $14 \times 14 \times 128$ |
| up0 | upsampling | - | - | - | None | $28 \times 28 \times 128$ |
| Conv5 | Convolutional | $3 \times 3$ | - | RELU | None | $28 \times 28 \times 128$ |
| Conv6 | Convolutional | $3 \times 3$ | - | RELU | None | $28 \times 28 \times 128$ |
| OutputT | Convolutional | $3 \times 3$ | - | RELU | None | $28 \times 28 \times 3$ |

We present the architecture of the network $A$ for the fair auto-encoder

| ID | Layer | Kernel | Stride | Activation | Normalization | Output |
|---|---|---|---|---|---|---|
| Input0 | Input | - | - | None | None | $28 \times 28 \times 3$ |
| Conv0 | Convolutional | $3 \times 3$ | - | RELU | spectral | $28 \times 28 \times 32$ |
| pool0 | max pooling | - | - | - | spectral | $14 \times 14 \times 32$ |
| Conv1 | Convolutional | $3 \times 3$ | - | RELU | spectral | $14 \times 14 \times 64$ |
| Conv2 | Convolutional | $3 \times 3$ | - | RELU | spectral | $14 \times 14 \times 128$ |
| pool2 | max pooling | - | - | - | spectral | $7 \times 7 \times 128$ |
| Flat 0 | flatten | - | - | - | spectral | 6272 |
| dense2 | dense | - | - | RELU | spectral | $32 \times 1$ |
| OutputA | Dense | - | - | linear | spectral | 1 |

