# OpenReview forum: "Fairness with Wasserstein Adversarial Networks"
_ICLR.cc/2020/Conference — Reject_

### Official Review · AnonReviewer1 · 2019-10-22
**Official Blind Review #1**

**Rating:** 1

**Review:**

In this paper, the authors proposed a fairness-aware learning method.
In particular, the authors considered two kinds of fairness problem and designed two regularizers accordingly.
Essentially, both of these two strategies learn classifiers and calibrate the distributions conditioned on protected variables jointly.
The calibration of the distribution is achieved in the framework of optimal transport.

This work is a natural extension of the optimal transport-based method shown in (Barrio et al, 2019a,b). The main differences include 1) instead of calibrating distributions after learning classifiers, the proposed method achieves calibration and learning jointly, replacing the primal Wasserstein barycenter problem with the dual form of Wasserstein distance (Arjovsky et al. 2017); 2) the proposed method considers two types of fairness problem.

Compared with vanilla GAN, the potential advantage of WGAN on distribution matching is well-known. It seems unfair that the authors compared the vanilla GAN-based regularizer with the proposed WGAN-based regularizer just on EMD because EMD corresponds to the proposed regularizer directly. In Table 1, although the DI of vanilla GAN is higher than that of WGAN, its ACC is also higher than that of WGAN as well. In Figure 5 (a, b), if we set lambda=0.6 for WGAN and lambda=1 for vanilla GAN, both of them can achieve ~0.838 ACC and ~0.100 DI. In Figure 5(c), what do the points represent? Why not use DI as the x-axis? Because of the issues in experiments, it is hard to evaluate the improvements of the proposed method.

Additionally, the proposed method always causes the degradation of ACC when improving DI. However, the method in (Barrio et al, 2019a) just applies a Wasserstein barycenter-based post-processing but can suppress the degradation on ACC greatly. Could the authors discuss the differences and the advantages of the proposed method in detail?  Could the authors consider more recent work as their baselines?

In summary, the method makes sense, but its novelty is limited and the improvements are incremental.

Minors:
Page 6, Line 3: Figure 3 —> Figure 2.
I suggest swapping Figure 2 and Figure 3.

**Experience Assessment:**

I have read many papers in this area.

**Review Assessment: Checking Correctness Of Derivations And Theory:**

I carefully checked the derivations and theory.

**Review Assessment: Checking Correctness Of Experiments:**

I carefully checked the experiments.

**Review Assessment: Thoroughness In Paper Reading:**

I read the paper thoroughly.

---

### Official Review · AnonReviewer2 · 2019-10-23
**Official Blind Review #2**

**Rating:** 1

**Review:**

The authors propose a method for adding an approximate disparate impact loss to a classification objective, and show that optimizing a classifier for this loss leads to "fairer" predictions with little or no accuracy loss.

The authors first formulate two notions of fairness in terms of earth mover distance between the distribution of scores conditioned on either value of a protected variable. They then show that the dual formulation of the earth mover distance can be approximated (specifically, lower-bounded) by optimizing the parameters of a neural network under spectral norm constraints. This leads to a min-max global optimization scheme to learn a fair classifier.

Strengths: The proposed method does better on the considered fairness metric than a GAN model with similar accuracy.

Weaknesses: The paper is difficult to read, glosses over some important details, and contains some inaccuracies.

-- Clarity:
--- The authors need to better describe the assumptions (or lack thereof) made on the joint distribution of X, S, and Y.
--- Measures such as the quantiles or probability laws need to be formally defined before they are used in definitions.
--- The \mathcal{L} notation is overloaded (it is used for probability laws, conditional and unconditional, as well as marginals, with \mathcal{L}_1 referring to both!), leading to potential confusion.
--- The domain of X and Y in equation (2) is not defined anywhere, neither is the distance.
--- Similar lack of consistency with the use of F / \mathcal{F} / \hat{f}, without any explicit parameterization
--- Figure 2 needs to be in Section 4, and Table 1 needs to be trimmed to size

-- Overlooked problems:
--- In the dual formulation, the optimization is done over a sub-set of Lipschitz function, hence approximation of \mathcal{W} is a lower bound at every step. Minimizing a lower bound on a loss can be justified, but requires more discussion
--- The trade-off inherent in the choice of n_w in algorithm 1 needs to be further discussed, especially in the case of large datasets where a full epoch of SGD in the inner loop of the optimization process is impractical

-- Inaccuracies:
The graphical models in Figures 1 and 2 and conditional independences written in the text are not consistent:
--- In Figure 1,  X is NOT independent of S given Y (neither is Y*) (see: V structures in a directed graphical model)
--- In Figure 2, X* is independent of S regardless of conditioning on Y

Considering all of the above issues, the paper is not currently ready for publication

**Experience Assessment:**

I have read many papers in this area.

**Review Assessment: Checking Correctness Of Derivations And Theory:**

I assessed the sensibility of the derivations and theory.

**Review Assessment: Checking Correctness Of Experiments:**

I assessed the sensibility of the experiments.

**Review Assessment: Thoroughness In Paper Reading:**

I read the paper thoroughly.

---

### Official Review · AnonReviewer3 · 2019-10-29
**Official Blind Review #3**

**Rating:** 1

**Review:**

This paper proposes a variant of adversarial learning to achieve some of the popular group fairness definitions. The main novelty is the idea of minimizing Wasserstein distance between the conditional distributions of classifier predictions given different values of the protected attribute.

My main concern is the approximation of a simple 1d Wasserstein distance with a neural network. Wasserstein distance between two discrete distributions in 1d can be computed in closed form (simple function of order statistics). That is, eq. (1) is simple to evaluate for two empirical distributions. There is no need to use a neural network for approximation, and even if authors choose to do so, some discussion on how well it approximates actual Wasserstein distance is needed. I think the proposed algorithm could be more interesting if authors can work out the optimization problem with the actual Wasserstein distance.

On the theoretical/motivation side, it is not enough to say that demographic parity is achieved when the corresponding Wasserstein distance is 0. What is needed is that demographic parity difference is bounded from above by the corresponding Wasserstein distance (I don't know if it is true or not, but would like to know). Then minimizing Wasserstein distance to achieve demographic parity could be justified.

Finally, the paper is quite poorly written. The description of fairness in the introduction is very vague. Authors essentially describe demographic parity as fairness, while it is simply one of the several definitions of group fairness. There is also individual fairness (the paper by Dwork et al. is cited, but not properly discussed) and prior work emphasizing certain deficiencies of group fairness [1] along with several recent papers studying individual fairness [2,3], some also utilizing Wasserstein distance [4].
Authors also provided incorrect definition of disparate impact. Equation in the bottom of page 2 corresponds to statistical parity difference, while disparate impact is the ratio.
"Equality of opportunity" on the top of page 3 seems to be a typo
"the mathematical properties of the disparate impact measure are not favorable, in particular it lacks robustness and smoothness features which would be necessary to blend algorithmic practice and mathematical theory" - I don't think this claim makes sense. There are many prior works studying disparate impact and proposing algorithms to achieve it, e.g. the cited work of Feldman et al. Authors should be more specific regarding what mathematical properties they consider not favorable.

There are a lot of typos and grammatical mistakes, e.g.
in the 1st paragraph of section 2.2, the sentence "Hence the aim in this case is to” is unfinished.
in the 1st paragraph of section 3, the first sentence seems to be unfinished.

[1] Kleinberg, J., Mullainathan, S., & Raghavan, M. (2016). Inherent trade-offs in the fair determination of risk scores.
[2] Kearns, M., Roth, A., & Sharifi-Malvajerdi, S. (2019). Average Individual Fairness: Algorithms, Generalization and Experiments.
[3] Jung, C., Kearns, M., Neel, S., Roth, A., Stapleton, L., & Wu, Z. S. (2019). Eliciting and Enforcing Subjective Individual Fairness.
[4] Yurochkin, M., Bower, A., & Sun, Y. (2019). Learning fair predictors with Sensitive Subspace Robustness.

**Experience Assessment:**

I have published one or two papers in this area.

**Review Assessment: Checking Correctness Of Derivations And Theory:**

I assessed the sensibility of the derivations and theory.

**Review Assessment: Checking Correctness Of Experiments:**

I assessed the sensibility of the experiments.

**Review Assessment: Thoroughness In Paper Reading:**

I read the paper at least twice and used my best judgement in assessing the paper.

---

### Decision · Program_Chairs · 2019-12-19

**Decision:**

Reject

**Comment:**

This paper presents an approach to enforce statistical fairness notions using adversarial networks. The reviewers point out several issues of the paper, including 1) their approach does not provably enforce criteria such as demographic parity, 2) lack of novelty and 3) poor presentation.